# Mapping Forest Abrupt Disturbance Events in Southeastern China—Comparisons and Tradeoffs of Landsat Time Series Analysis Algorithms

Ning Ding [1,2] and Mingshi Li [1,2,*]

1   Co-Innovation Center for Sustainable Forestry in Southern China, Nanjing Forestry University, Nanjing 210037, China; dingn1023@njfu.edu.cn
2   College of Forestry, Nanjing Forestry University, Nanjing 210037, China
*   Correspondence: nfulms@njfu.edu.cn; Tel.: +86-2585427327

**Abstract:** Forest change monitoring is a fundamental and routine task for forest survey and planning departments, and the resulting forest change information acts as an underlying asset for sustainable forest management strategy development, ecological quality assessment, and carbon cycle research. The traditional ground-based manual monitoring of forest change has the disadvantages of high time and labor costs, low accessibility, and poor timeliness over wide regions. Remote sensing technology has become a popular approach for multi-scale forest change monitoring due to its multiple available spatial, spectral, temporal, and radiometric resolutions and wide coverage. Particularly, the free access policy of long time series archive data of Landsat (around 50 years) has triggered many automated analysis algorithms for landscape-scale forest change analysis, such as VCT, LandTrendr, BFAST, and CCDC. These automated algorithms differ in their principles, parameter settings, execution complexity, and disturbance types to be detected. Thus, selecting a suitable algorithm to satisfy the particular forest management demands is an urgent and challenging task for forest managers in a given forested area. In this study, Lishui City, Zhejiang Province, a typical plantation forest region in Southern China where forest disturbance widely and frequently exists, was selected as the study area. Based on the GEE platform, the algorithmic adaptability of VCT, LandTrendr, and CCDC in monitoring abrupt forest disturbance events was compared and assessed. The results showed that the kappa coefficients of the abrupt disturbance events detected by the three algorithms were at 0.704 (LandTrendr), 0.660 (VCT), and 0.727 (CCDC), and the corresponding overall accuracies were at 0.852, 0.830, and 0.862, respectively. The validated disturbance occurrence time consistency reached nearly 80% for the three algorithms. In light of the characteristics of forest disturbance occurrence in southeastern China as well as the algorithmic complexity, efficiency, and adaptability, LandTrendr was recommended as the most suitable one in this region or other similar regions. Overall, the forest change monitoring process based on GEE is becoming more simplified and easily implemented, and the comparisons and tradeoffs in this study provide a reference for the choice of long time series forest monitoring algorithms.

**Keywords:** LandTrendr; VCT; CCDC; forest disturbance; GEE



## 1. Introduction

Changes in forest ecosystems have important and far-reaching implications for both human and biosphere conditions as a whole [1]. Forest change monitoring is the primary means to understand the dynamics of forest ecosystem and forest resource development trends, enabling the guidance of forest management, planning, and restoration activities. Particularly, the resulting spatio-temporally explicit forest change information from the monitoring is crucial to the strategic development of sustainable forest management and forestry industry planning and upgrading [2]. Ecologically, a forest disturbance (change event) is defined as a temporary change in the state of the forest environment, which can

bring about significant changes in the forest ecosystem's structure and function, such as a substantial biomass loss [3]. Forest disturbances are caused by human activities, such as logging and forest thinning, or natural factors, such as landslides and sudden climate changes. Overall, these forces-induced forest change can be categorized into four categories when using remote sensing data to detect [4]: (1) abrupt change, (2) seasonal change, (3) gradual ecosystem change, and (4) short-term inconsequential change, of which, from the perspective of restoration ecology, abrupt change is the most significant type of forest change in forest management, and it is most easily detected by comparing multi-temporal remote sensing observations. Thus, this study focuses on abrupt change events to test the adaptability of different automated analysis algorithms based on Landsat time series observations.

Traditional remote sensing-based forest disturbance or change mapping is principally based on comparing bi-temporal or multi-temporal satellite scenes, followed by the analysis of forest cover status or the calculation of increase or decrease in relevant indices to derive a general picture of forest changes [5–7]. Obviously, this manner is inefficient in the change analysis of long time span, e.g., 1990 to 2020, and requires the analysts to be well-trained and have rich experiences in remotely sensed image analysis. With Landsat opening all its historical archive data to the public for free access, monitoring forest change over a long time span by using consecutive years' images has now been the norm [8]. Based on the dense image stacks, long-term forest change monitoring algorithms built upon spectral variable tracking, image classification, spectral trajectories-based analysis, data fusion, and other different means have surged [9]. The most reputational automated algorithms for forest change analysis include the vegetation change tracker (VCT) model [10], the Break detection For Additive Season and Trend (BFAST) [11], the Landsat-based detection of Trends in Disturbance and Recovery (LandTrendr) [12,13], and the Continuous Change Detection and Classification (CCDC) [14]. In recent decades, the development and promotion of big data platforms have broken through the limitations of storage and computing power, for example, the combination of big data and cloud computing on Google Earth Engine (GEE) [15] makes these automated forest change analysis algorithms easily and efficiently implemented in GEE environment, popularizing their practicability in more fields [16–18]. However, some algorithm application problems have also arisen with the popularization, such as the incomplete matching of algorithm monitoring advantages with the actual forest management demands [19], applications based solely on algorithm availability [20–22], and the inability to perform algorithm selection with generalizable quantitative criteria and evaluation systems [8,23]. Therefore, it is necessary to compare the effectiveness and adaptability of the automated algorithms based on the GEE platform to guide a proper selection of algorithms in particular forested areas to address these concerns.

Lishui has the highest forest coverage in Zhejiang Province and is a pioneer in China's forestry industry development [24,25]. The area of Lishui is characterized by extensive plantation forests, and regular forest logging and post-harvesting forest recovery practices, as well as casual forest fire events, are frequently witnessed in its forest ecosystems [26]. Thus, Lishui can be considered a typical prototype area to test the effectiveness of diverse automated time series analysis algorithms. Moreover, the anthropogenic disturbance type is more prominent in Lishui's forest, which is a common feature of the forests in southeastern China, helping to verify the transferability of the ultimately selected algorithm.

In summary, based on the long-term research and monitoring needs for the forests in southeastern China, it is necessary to select an adaptable and efficient long time series forest monitoring method for this region. The major objective of this study was to identify the most suitable automated time series analysis algorithm for the forests in Southeast China from GEE platform by considering algorithmic complexity, efficiency, and the match degree of the detected disturbance types to the purposes of forest management.

## 2. Study Area and Materials

The study area, Lishui (118°41′–120°26′E, 27°25′–28°57′N), is located in the southwestern part of Zhejiang Province (Figure 1). Lishui is a mountainous city with good hydrothermal conditions, which are conducive to vegetation growth. The main dominant species are masson pine (*Pinus massoniana* Lamb.), Chinese fir (*Cunninghamia lanceolata* (Lamb.) Hook.), and oak (*Cyclobalanopsis glauca* (Thunb.) Oerst.), and they constitute over 70% of the area of forested land in Lishui [27]. Lishui belongs to the subtropical monsoon climate zone, with more obvious subtropical maritime monsoon climate characteristics. In recent decades, forest change events have been frequent in Lishui, mainly caused by forest harvesting, pests and diseases, and post-disturbance reforestation [26].

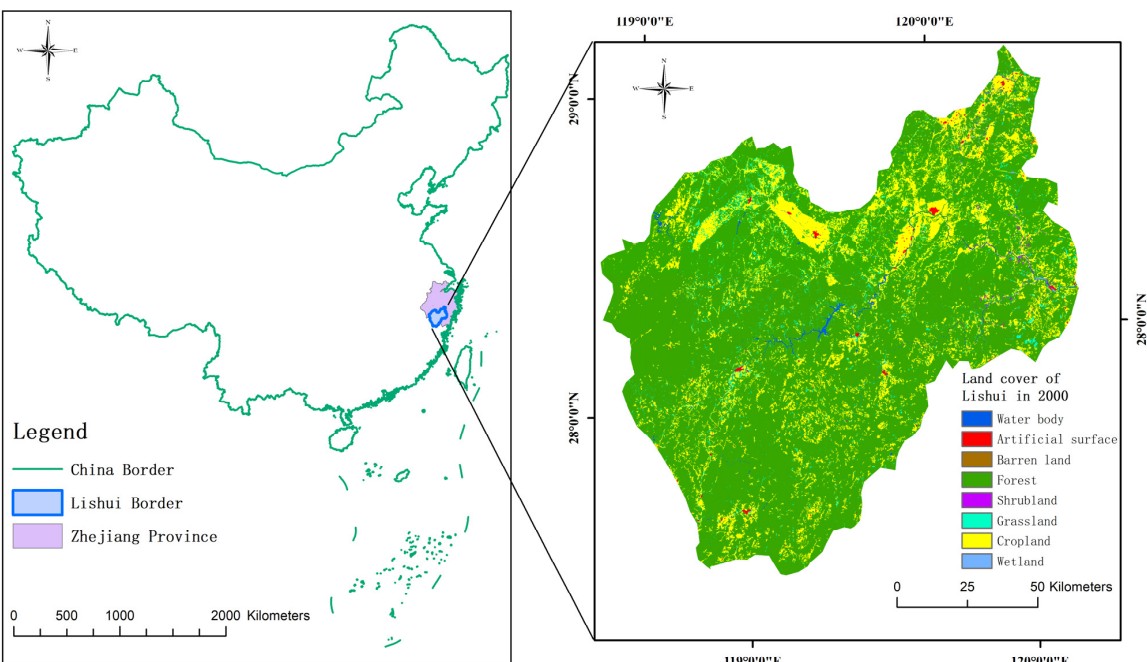

**Figure 1.** Maps showing the location of the study area and land cover types in 2000 at Lishui City, Zhejiang Province, China. The right map is the 2000 Land cover data supporting VCT algorithm running, which is a subset of the GlobeLAND30.

In this study, the Landsat images archived in the GEE platform with a time span from 1990 to 2020 were used as the data source for monitoring those abrupt forest change events in Lishui City. Landsat images have the characteristics of free access, long observation history, and good spectral consistency among different sensors, including TM, ETM+, OLI, and OLI-2, which are very suitable for land cover change analysis, particularly the forest change analysis at landscape scale [28]. Using Landsat data archived by the GEE platform, the spectral characterization among several sensors was maintained [29–31], thus we did not need to perform additional pre-processing, such as radiometric normalization [32,33]. To guarantee the credibility of the detected abrupt forest change events from different automated analysis algorithms, we used the following image selection criteria: (1) cloud-free or less than 10% cloud coverage, and (2) the image acquisition date falling into mid-June through mid-September for mid-altitude region to make sure the vegetation growth is vigorous, to select Landsat images to facilitate the running of VCT and LandTrendr algorithms. For CCDC algorithm, we used all the archived images in the GEE platform from 1990 to 2020 to fit a full model of CCDC to detect the abrupt forest change events. Additionally, due to the widespread stripe errors in the Landsat 7 images after May 2003, we used the temporally corresponding Landsat 5 TM images as substitutes to support the running of all three detection algorithms. Moreover, it is worth noting that the official Chinese GlobeLAND30 global geographic information public product (http://www.globallandcover.com/home.html ac-

cessed on 22 February 2022) was used to streamline the running of VCT algorithm. We used PyCharm Community Edition 2023.1.2 and ArcMap 10.8 to perform the necessary data batch processing and analytical mapping operations.

The validation data used in this study was the archived high spatial resolution Google Earth Maps resided in Google Earth Pro, which were visually interpreted as the reference data to validate the accuracy of the three automated algorithms. When the high-resolution maps were unavailable in certain years, we had to visually interpret corresponding Landsat image pairs directly to produce the reference data.

## 3. Method

In this study, the vegetation change tracking algorithm (VCT) [10], LandTrendr algorithm [12,13], and continuous change detection and classification algorithm (CCDC) [14] were compared in mapping abrupt forest disturbance events under the same conditions to identify the most suitable algorithm.

The VCT algorithm calculates the normalized integrated forest index (IFZ) and sets the change threshold according to the actual forest coverage situations to identify forest change [10]. IFZ is calculated from band spectral values, spectral value averages, and standard deviation scores, which can characterize the changes in forest pixels more comprehensively, as shown in Equations (1) and (2). Generally, VCT is capable of capturing those abrupt forest change events, such as clearcuts, fire-induced forest loss, and high-intensity thinning, and insensitive to those low-intensity forest canopy change events [10].

$$FZ_i = \frac{\left(b_{pi} - \overline{b_i}\right)}{SD_i} \tag{1}$$

$$IFZ = \sqrt{\frac{1}{N}\sum_{i=1}^{N}(FZ_i)^2} \tag{2}$$

where $b_{pi}$ is the reflectance of pixel p in band $i$, $\overline{b_i}$ and $SD_i$ are the mean and standard deviation of the reflectance of band $i$, $N$ is the number of bands involved in the calculation.

For annual Landsat TM and ETM+ images, only band 3 (Red band, R), band 5 (Short-wave Infrared 1 band, SWIR1), and band 7 (Short-wave Infrared 2 band, SWIR2) are used for the calculation [10]. For Landsat OLI and OLI-2 images, the spectrally corresponding bands are band 4 (R), band 6 (SWIR1), and band 7 (SWIR2) for the calculation, respectively. After the classical desktop stand-alone version of VCT algorithm is introduced into GEE platform, the difficulty of acquiring high-quality images and significant computational cost is accordingly solved to the greatest extent possible [17]. Thus, combined with high-quality land cover data, VCT can be easily implemented in GEE environment to map yearly abrupt forest disturbance events.

The LandTrendr algorithm extracts the spectral time series of forest pixels for trend fitting and breakpoint analysis. The abrupt change of forest pixels is recorded from the sudden spectral change in the time series, which corresponds to a spectral trajectory breakpoint to delineate the trend change of the forest [12]. The model can be constructed from simple to complex and from rough to smooth to achieve the purpose of eliminating noise and streamlining details, as shown in Figure 2. It is important to note that most of the changes monitored as trends over a long period are not caused by noise, but the changes within a year are more likely to be caused by spectral noise. Therefore, in the GEE version, the parameters of LandTrendr are set to treat the changes within a year as errors, and the GEE platform also simplifies the pre-processing steps of LandTrendr by presenting the core time series segmentation algorithm in its entirety and even improving it [16]. The GEE version of LandTrendr algorithm eliminates restrictions on raw data and computational costs, reducing the barriers to applying the algorithm and allowing more non-professionals to use LandTrendr algorithm to make breakthroughs in their own areas of expertise. The

current work made adjustments to each of the LandTrendr parameters suitable for the natural conditions of Lishui City, to improve the spatial and temporal accuracy of the abrupt forest change monitoring.

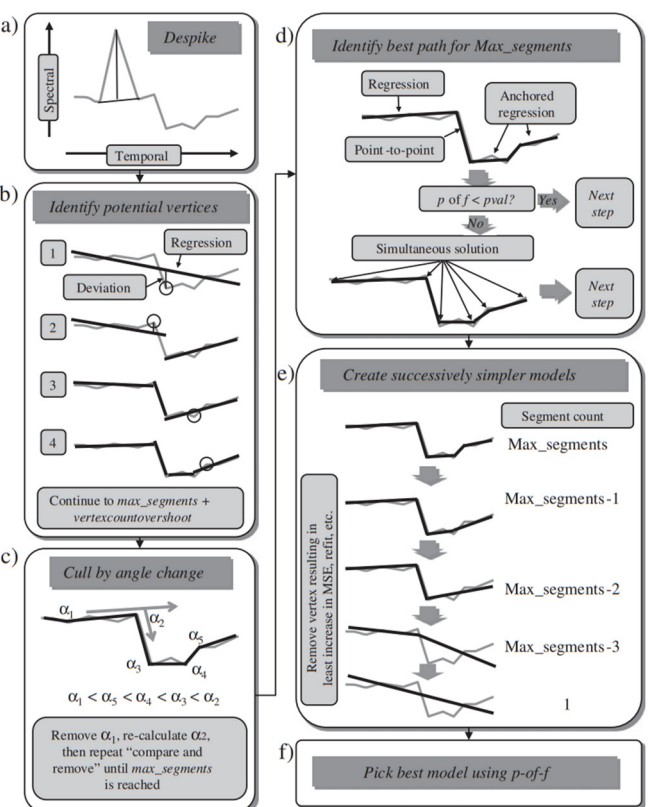

**Figure 2.** The conceptual diagram of LandTrendr change detection algorithm [12]. After removal of ephemeral spikes (**a**), potential vertices are identified using deviation from simple regression lines (**b**) as much as possible. Then, excess vertices are removed based on low angle change (**c**), according to the max_segments, a single path of model was chosen (**d**) and creating successively simplified models of the trajectory with removing segments (**e**). The model with the best fit is chosen (**f**).

The image processing for local application of CCDC algorithm is very costly, which necessitates the development of an online version of CCDC in the GEE platform. The core idea of CCDC is to build a time series model for each pixel in each band to monitor trend changes, seasonal variations, and abrupt changes in forest landscapes following the Fourier harmonic model thoughts [34]. The improved advanced CCDC model adds two intra-annual bimodal variation coefficients, and the full model of CCDC adds a pair of intra-annual trimodal variation coefficients based on the advanced CCDC model, as shown in Equation (3), to enhance the modeling performance of the intra-annual variation of the Landsat time series [14]. The model recalculates the coefficients once a new image is added to the time series, so the model refitting is actually real-time with newly added image data, which is a unique feature of model building known as "online" [35]. Once the newly added image spectral value differs substantially from the previous model fitting value, for example, the difference between them exceeding triple times of RMSE, the new image data are considered to provide an "abrupt change" for the time series and the mutation in spectral value ends the previous model and restarts the next modeling process [34].

$$\hat{\rho}(i, x) = a_{0,i} + \sum_{k=1}^{n} \left( a_{k,i} cos\left( \frac{2k\pi}{T} x \right) + b_{k,i} sin\left( \frac{2k\pi}{T} x \right) \right) + c_{1,i} x \qquad (3)$$

where $\hat{\rho}(i, x)$ is the predicted spectral reflectance of the $i$th band for the $x$th Julian day of the fit, $i$ represents the $i$th spectral band, and $T$ represents the number of days per

year, $a_{0,i}$ represents the annual overall coefficient of the *i*th band, and $c_{1,i}$ represents the annual coefficient of variation of the *i*th band. The $a_{k,i}$ with $b_{k,i}$ represents the intra-annual coefficient of variation of reflectance of band *i*, the *k* represents the pairs of the intra-annual coefficients (such as $a_{1,i}$ and $b_{1,i}$), and in the complete CCDC fitting model, $k = n = 3$, which means three pairs of the intra-annual coefficients will be used.

Of course, accurate fitting requires numerous consecutive clear images as the basis. When the number of consecutive clear images is greater than or equal to 6, the simple model of CCDC can be fitted, and when the number of consecutive clear images exceeds 24, the full CCDC model is estimated by using the LASSO modeling method [14]. The clearer observations, the more complex the model used, and the better the fitting effect. Figure 3 shows the model fitting differences among the three-level CCDC models. When porting CCDC to GEE platform, the algorithm breaks through the application space limitation, allowing users to apply the algorithm globally to facilitate their own research [14].

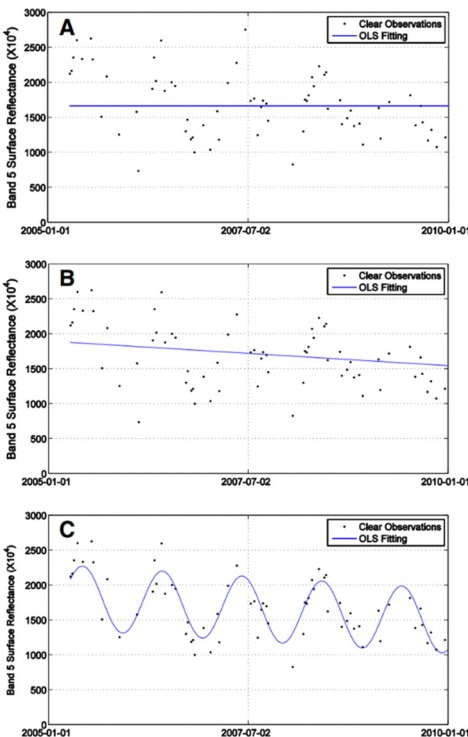

**Figure 3.** Schematic diagram of CCDC fitting model [34]. (**A**) shows the results of a simple model fitting, (**B**) shows an advanced model fitting, and (**C**) shows the full model fitting to derive three sets of coefficients.

At this stage, the main approach for forest change detection results validation is still to build a confusion matrix to derive related statistics [8,20]. Building a confusion matrix can unify different detection algorithms into the same validation framework, making the accuracy statistics comparable with each other. In this study, the results obtained by the three algorithms were verified from two aspects: temporal and spatial accuracy verification. In the validation of algorithmic temporal accuracy, 1550 random points were generated uniformly by year via implementing a stratified random sampling in the change areas detected by the three algorithms. Visual interpretation of the reference image pairs containing these random points was performed to count their real change number to derive the accuracy statistics. It is necessary to note that those random points when their detected change time was different from the actual change time by more than 5 years were considered false detections. After weighing similar studies [21,36] and the general conditions in Lishui [26], we argued that it was feasible to set the time bias to 5 years. If the interval between the disturbed time detected by the algorithms and the actual occurrence time of disturbance was more than 5 years,

it was considered to be two different disturbance events at the same location. Similarly, a stratified sampling method was used to verify the spatial accuracy, in which we selected 500 random points in each area identified as changed class or unchanged class by each algorithm in order to obtain an unbiased accuracy estimation.

In terms of the algorithmic performance evaluation, this study evaluated various aspects, including the spatial and temporal accuracy, model principle differences, disturbance types to be detected, time and space costs of running the algorithms, and the limitation of the algorithms to guide the selection of a more suitable long time series monitoring algorithm in southeastern China.

## 4. Results

### 4.1. Spatial Accuracy of the Detected Disturbance Events

After visually interpreting the corresponding Google Earth maps or original Landsat image pairs as the reference data, the accuracy statistics, including the overall spatial accuracy (OA), kappa coefficients, user's accuracy, and producer's accuracy of LandTrendr, VCT, and CCDC detected forest disturbance results were derived and summarized in Table 1, Table 2, and Table 3, respectively.

**Table 1.** The validation accuracy statistics of LandTrendr-detected abrupt forest disturbances.

| LandTrendr | Reference Data | | | |
|---|---|---|---|---|
| | Forest Change | Non-Change | Total | User's Accuracy |
| Forest change | 466 | 34 | 500 | 0.932 |
| Non-change | 114 | 386 | 500 | 0.772 |
| Total | 580 | 420 | 1000 | |
| Producer's accuracy | 0.803 | 0.919 | | |
| | OA: | 0.852 | Kappa: | 0.704 |

**Table 2.** The validation accuracy statistics of VCT-detected abrupt forest disturbances.

| VCT | Reference Data | | | |
|---|---|---|---|---|
| | Forest Change | Non-Change | Total | User's Accuracy |
| Forest change | 448 | 52 | 500 | 0.896 |
| Non-change | 108 | 392 | 500 | 0.784 |
| Total | 556 | 444 | 1000 | |
| Producer's accuracy | 0.806 | 0.883 | | |
| | OA: | 0.840 | Kappa: | 0.680 |

Comparing Tables 1–3 shows that LandTrendr had the highest user's accuracy at 0.932 and a producer's accuracy of 0.803 for the forest disturbance class, and the least omission error of 0.081 (1.000–0.919) for the non-change class among the three algorithms. On the whole, LandTrendr algorithm gained an OA of 0.852, with a kappa coefficient at 0.704, indicating that LandTrendr missed certain real forest changes, but more than 90% of the detected forest changes were correct. Additionally, a certain degree of omissions occurred in LandTrendr's statistical data (the producer's accuracy of 0.803 in forest change class) caused by LandTrendr parameter settings, where some short-term and less intense disturbance events were discarded in the detection process of LandTrendr. Although the stripe errors of Landsat 7 were eliminated, the high-quality demand for the input Landsat images made the monitoring performance of VCT limited by the number of available images. Table 2 shows that the OA (0.840) and kappa coefficient (0.680) of VCT were not very high, and the user's accuracy (0.896) and producer's accuracy (0.806) suggested that VCT remained sensitive in monitoring abrupt changes. CCDC achieved a more balanced effect in the two accuracy directions of user's (0.898) and producer's (0.838), with the highest OA (0.862) and kappa

coefficient (0.724) and the least commission error of 0.174 (1.000–0.826) for the non-change class among the three algorithms (Table 3). Under the same evaluation framework, CCDC performed best on most of the evaluation indices.

**Table 3.** The validation accuracy statistics of CCDC-detected abrupt forest disturbances.

| CCDC | Reference Data | | | |
|---|---|---|---|---|
| | **Forest Change** | **Non-Change** | **Total** | **User's Accuracy** |
| Forest change | 449 | 51 | 500 | 0.898 |
| Non-change | 87 | 413 | 500 | 0.826 |
| Total | 536 | 454 | 1000 | |
| Producer's accuracy | 0.838 | 0.890 | | |
| | OA: | 0.862 | Kappa: | 0.724 |

### 4.2. Temporal Accuracy of the Detected Disturbance Events

Figure 4 conveys the temporal accuracy dynamics of the three algorithms detected abrupt forest disturbance events. Overall, the temporal accuracies of CCDC and LandTrendr were better than that of VCT, with their average temporal accuracy reaching 87.30% (CCDC), 78.97% (VCT), and 87.74% (LandTrendr). Since LandTrendr and CCDC constructed time series analysis models based on the input images, which enabled the fitting of the analysis models to include the year 2012 for disturbance detection mapping, but this inclusion was not the case for VCT due to the absence of the 2012 images (abandoning the Landsat 7 ETM+ image due to its stripe errors), thus, the monitoring in 2012 was skipped in the time series analysis of VCT (Figure 4). The temporal accuracy of VCT fluctuated more intensely year by year, and the accuracy of VCT depended only on the quality of the respective selected images in each single year and did not affect the accuracy in nearby years. In contrast, LandTrendr and CCDC, which needed to build time series analysis models for forest disturbance monitoring, had different degrees of accuracy reduction at the beginning and end of the time series (Figure 4).

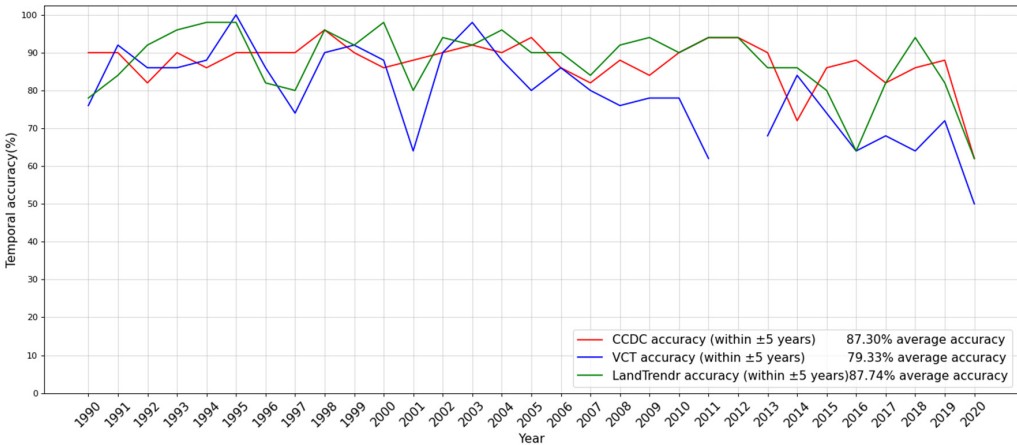

**Figure 4.** The validated temporal accuracy dynamics of LandTrendr, VCT, and CCDC detected abrupt forest change events.

Figure 5 shows the wall-to-wall maps of the abrupt forest disturbance events during the period 1990 to 2020 in Lishui City, detected by VCT, LandTrendr, and CCDC, respectively. Overall, the spatial distribution patterns of the disturbances mapped by the three algorithms were similar, but the disturbed patches detected by LandTrendr and CCDC were much less than those of VCT.

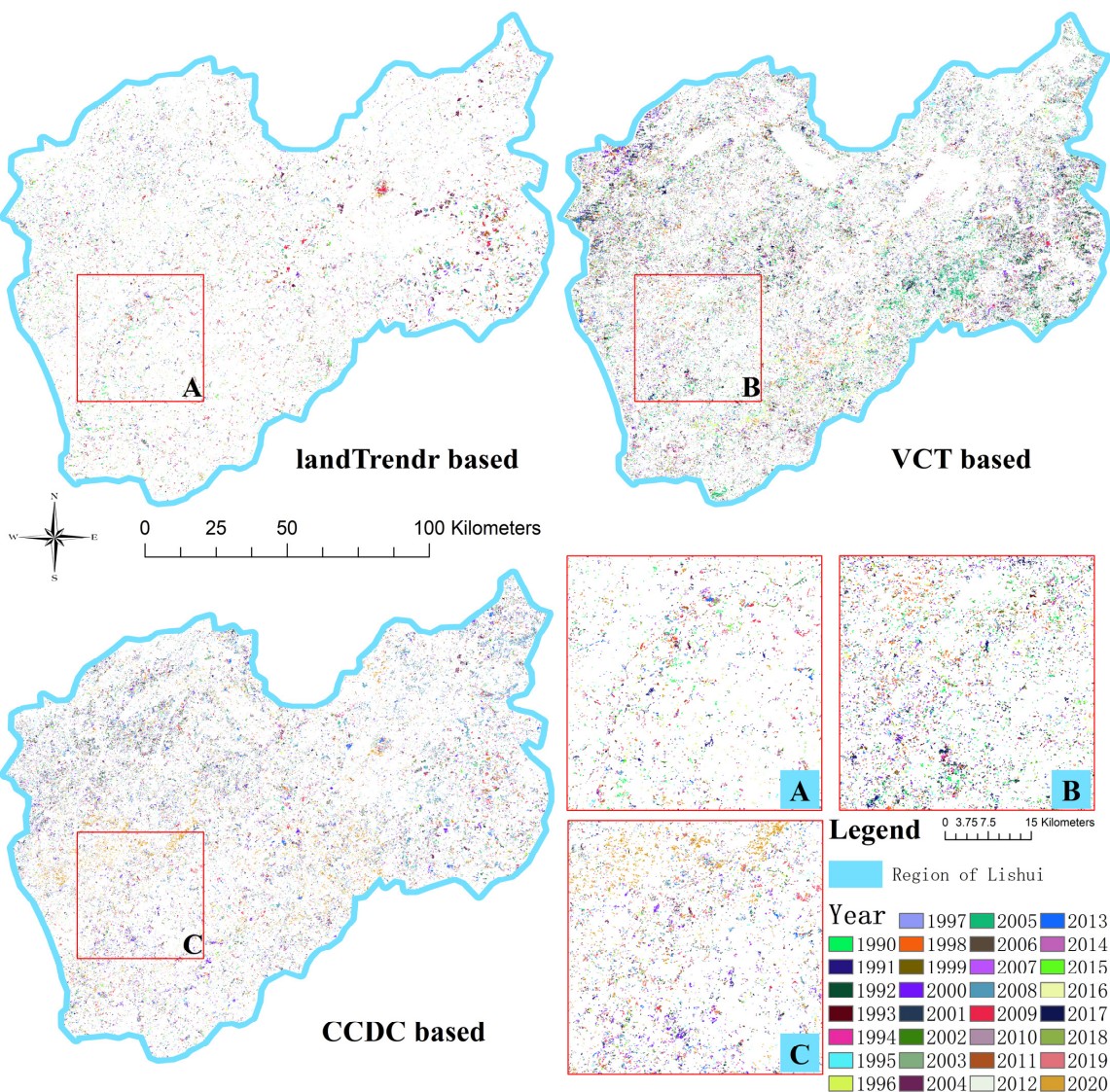

**Figure 5.** Forest abrupt changes mapping from 1990 to 2020 in Lishui derived from LandTrendr (**A**), VCT (**B**), and CCDC (**C**) algorithms.

### 4.3. Field Patch Matching

For the same forest change events on the ground, algorithms may give different detection results in terms of temporal (the timing of forest change) and spatial (the morphology of forest change patches) features. The number and morphology of the disturbance patches obtained by each algorithm were different. In terms of numbers, LandTrendr had the smallest number of patches at 32,827, CCDC had a total of 65,011, and VCT obtained the largest number of patches at 94,547. The number differences in the disturbance patches detected by the three algorithms were attributed to the algorithmic principle differences. Compared to CCDC and LandTrendr, VCT did not construct a time series model, so it was much rougher in change monitoring, with more fragmented changes and more patches. LandTrendr and CCDC constructed time series analysis models for pixels to portray the pixel change process. CCDC retained the complete time series change simulation, but LandTrendr performed model simplification (Figure 2e) and change filtering to streamline the model, therefore, LandTrendr obtained the fewest change patches.

Figure 6 exhibits an example of how progressive deforestation events were captured by the three algorithms. Obviously, the morphologies of patches detected by LandTrendr and CCDC in different years were more regular and consistent with the actual shapes

depicted by corresponding high-resolution Google Earth maps and Landsat 8 data than VCT-depicted morphology, and VCT and CCDC-detected disturbance patches were more finely fragmented than LandTrendr detected patches. VCT did not construct a time series, hence, its patches were more disorganized, while CCDC monitored more sub-year changes of vegetation, so the patches of disturbance occurring at different times were more detailed. In terms of the general morphologies of patches, LandTrendr's result was accurate and clear, but VCT over-depicted the number of disturbance patches and CCDC fragmented the patches due to over-sensitivity (Figure 6).

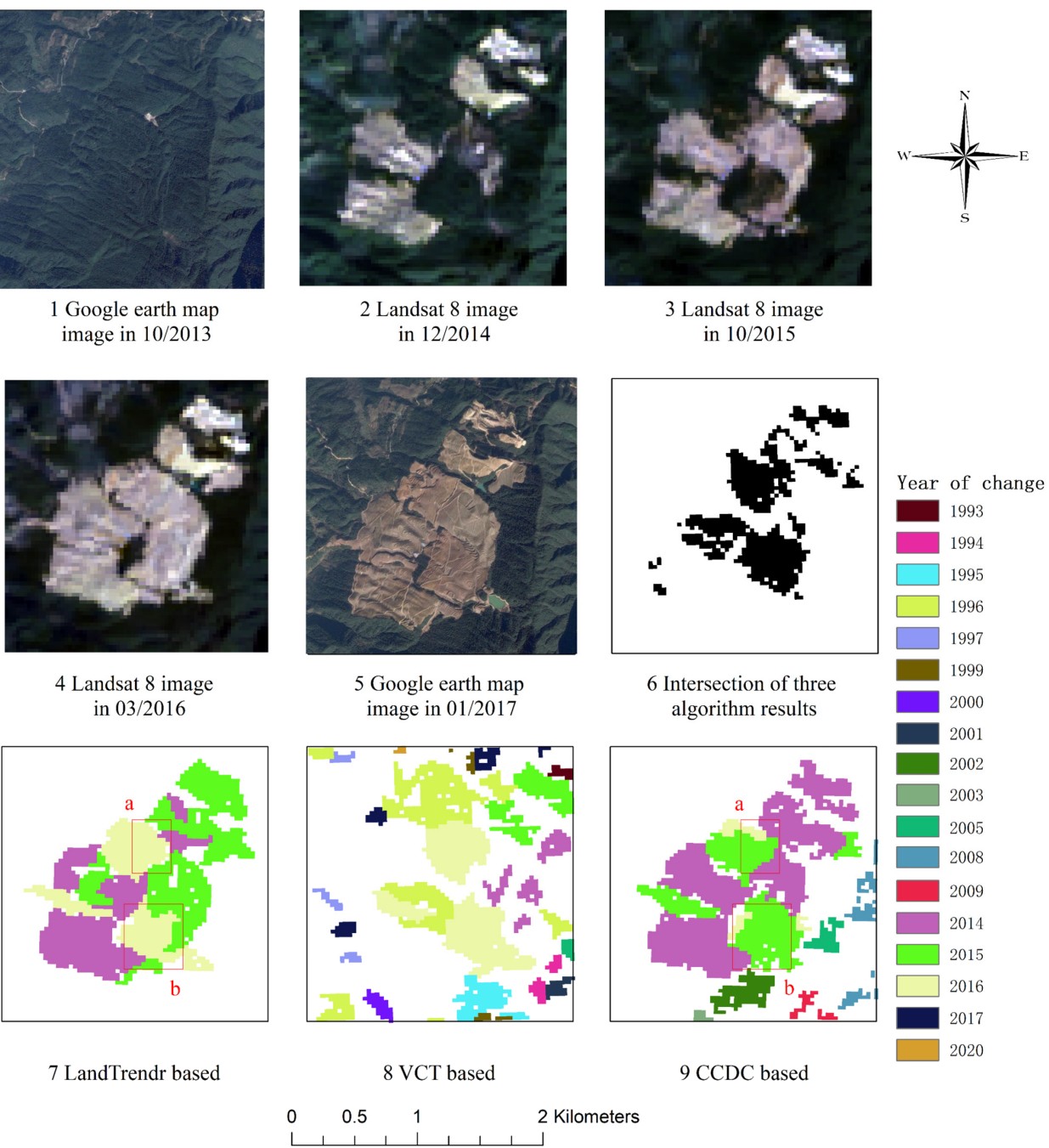

**Figure 6.** The validation of the progressive abrupt disturbance events detected by VCT, LandTrendr, and CCDC by using high spatial resolution Google Earth Maps and Landsat images. Locations labeled as a and b show the difference between the responses of Landtrendr and CCDC to the time of change.

From the aspect of monitoring the disturbance occurrence time, Figure 6 shows that CCDC-monitored change time was clearer than LandTrendr and VCT. We found that some abrupt changes which occurred in late autumn and winter months of the previous year might be mapped into the next year in LandTrendr's monitoring results. LandTrendr was a bit correctly monitored for changes that occurred in 2014, but there were also some changes that were relegated to 2015 (Figure 6). That was mainly because the images selected for the algorithm construction were narrowed in the vegetation growth period, thus, these selected images could not capture the spectral signals of forest changes occurring in the late autumn and winter months. In contrast, the change time monitoring in 2014 of CCDC was more detailed and accurate. For changes that occurred between October 2015 and March 2016, and their exact occurrence times were not given by the images, CCDC's monitoring results were earlier than LandTrendr overall (locations labeled as a and b in Figure 6). This is undoubtedly due to CCDC's more sophisticated modeling and richer data series used, but this delicacy is also making the monitored patches and timing of change fragmented.

## 5. Discussion

### 5.1. Comparative Evaluation of the Algorithms

Each of the three algorithms has its own advantages and disadvantages, and different situations need to be considered in the process of practical application to choose the appropriate method. Table 4 details the differences between the three GEE version algorithms.

**Table 4.** Trade-off factors for LandTrendr, VCT and CCDC in the implementation process.

| Terms | LandTrendr | VCT | CCDC |
|---|---|---|---|
| Source Online/offline | Online documentation with [16] Offline | Code cases with [17] Offline | Online Tools and [18] Online |
| Principle | Monitoring "vertices" based on time series, threshold determination changes | Monitoring forest change by calculating IFZ contrast thresholds | Constructing independent segments based on time series and calculating different model coefficients for each segment to record changes |
| Land cover type | all land cover changes | forest changes | all land cover changes |
| Composition of results | Changes are filtered by the platform, and the resultant data are directly exported for a total of 6 bands | Changes by year are exported by the platform, with subsequent filtering to synthesize outcome data | A total of 75 bands were exported by the platform for each individual segment parameter and related information slice file |
| Time of running | Within 1 h | Several hours | Several days |
| Selection of results | Output the patches after filtering | Output all patches yearly | Output all patches monitored |
| Images used | Using vegetation growth period images | Using vegetation growth period images | Using year-round images |
| Types of change | Abrupt and trend changes | Abrupt changes | Abrupt, trend, and gradual ecosystem changes |
| Scale of time | Interannual monitoring | Interannual monitoring | Sub-annual monitoring |

In terms of the complexity of model construction, it is progressively more complicated from VCT, LandTrendr, to CCDC. And all three algorithms are pixel-based change monitoring algorithms, enabling them to be validated by a unified standard. With the help of introducing a land cover classification map, VCT can distinguish forest pixels from the rest of other land cover pixels, thus, it can categorize forest pixels into persisting forests and non-persisting forests during the entire monitoring period. Such information that cannot be provided by the other two algorithms is just a unique output of VCT. Additionally, VCT has a high demand on the quality of the input Landsat images, e.g., cloud-free and images acquired in the peak season of plant growth, and the changeable threshold settings, e.g., IFZ

not less than 0.3 for dense forest coverage regions and IFZ not less than 0.2 for sparse forest coverage regions. However, due to the difficulty in achieving the high demands of VCT, some commission and omission errors take place in the monitoring process, leading to the relatively low accuracy of VCT [8]. LandTrendr also requires the input Landsat time series images falling into the peak growing season, the same as VCT, but the model of LandTrendr is constructed by associating images with individual years and describing the time series by "vertices" and "segments" [23]. CCDC is also the method for forest monitoring by constructing a time series model, which is based on annual data for model construction. However, compared to the model of LandTrendr, CCDC associates images with specific dates of image acquisition to construct "breakpoints" and independent segments [23]. Meanwhile, CCDC uses simulation to obtain Landsat data at any date [34], which greatly enriches the number of images involved in the monitoring. Because of the high precision of the pixel time series characterized by various coefficients, CCDC does not filter out its monitored changes directly, and analysts can choose the rules to keep and filter the disturbance results. The high level of specialization also results in significant computational and data storage costs. Performing CCDC on the GEE platform shifts the computational processing pressure to the cloud platform and outputs and stores the time-series coefficient data and disturbance information in the form of slices, but it is still tightly controlled to geographic extent.

During the implementation process, LandTrendr is more flexible and lenient for temporal screening of the input data, and its parameters and thresholds can be adjusted directly [16]. However, both VCT and CCDC encapsulate this section completely without requiring researchers to adjust themselves. The encapsulation of algorithms also limits the potential for the algorithms to be more regionally specific. The three algorithms have different complexities and take different amounts of time to obtain the results on the GEE platform. Taking the current work as an example, LandTrendr took about half an hour to obtain results, VCT took more than two hours, and the speed of CCDC getting data on the platform was related to the size and shape of the study area, and CCDC required researchers to filter changes monitored subsequently. As a result, CCDC took about seven days to achieve the forest disturbance distribution maps like LandTrendr's results.

The three algorithms also differ in the way they organize their results. VCT publishes the outcome data in the form of annual forest disturbance distribution maps within the entire monitoring period [17]. In addition to providing information on the timing of the disturbances, LandTrendr obtains information on the duration and spatial extent of the disturbances. LandTrendr organizes the results in terms of disturbance time, variability, duration, and other disturbance elements and filters the forest disturbances monitored in the same region at different times, only keeping the most intense disturbances in history [16]. CCDC does not do filtering directly, and it can obtain forest change status in any area at any date, but if we want to obtain the complete CCDC monitoring results of the whole area within the entire time period, we need to filter and collocate the result data.

In terms of forest disturbance types to be monitored, the three algorithms respond sensitively to abrupt changes occurring in the forest. Compared with VCT, which is good at capturing the abrupt forest changes year by year, CCDC and LandTrendr are more detailed in monitoring forest disturbances. The advantages of CCDC over LandTrendr are the sub-annual monitoring and the "online" modeling approach [35]. This is why CCDC can monitor the gradual changes in the forest. These advantages of CCDC were confirmed during the verification of the algorithm accuracy in Lishui, but some of the trend changes were overwritten by more obvious and drastic abrupt changes in the 31 years of monitoring. Therefore, although the advantages of CCDC were not fully exploited in this study, the temporal sensitivity of CCDC was substantially higher than the other two methods.

The three algorithms can theoretically carry out long time series of forest monitoring for any area, but each has some limitations in the specific operation process. The limitation of VCT is that it needs to input a rough land cover classification map of the monitoring area to drive the running of VCT. For areas without official land cover classification maps, ana-

lysts have to develop this rough map beforehand and convert or aggregate the classification codes in compliance with NLCD's definitions [17]. Meanwhile, the maximum monitoring period of the GEE platform version of VCT method is 30 years [17], which limits the application of VCT to longer time series. The limitation of LandTrendr is that the disturbance determination threshold is defined by the user, which can be more region-specific, but also makes the model structure uncontrollable under extreme conditions and causes the subjectivity of the results to some extent. CCDC may have errors when monitoring in areas with large inter-annual variability, so it is less effective when transplanting to semi-arid areas [34]. However, leaving aside the structural limitations, CCDC has no limitations in data acquisition and timing and is highly portable.

For the local forest disturbances in Lishui, three methods relied on their own focus to obtain reliable monitoring results. LandTrendr had the most refined results, captured the most intense and prominent disturbance patches, and neglected some small patches; VCT focused on the overall changes in the whole area, and some details might appear to be rough; CCDC produced reliable results, both for the abrupt changes that dominate the area and for those seasonal and gradual changes in the forest. In such an abrupt change pattern, CCDC was more accurate in monitoring the time of change patches and achieved the best spatial accuracy, but LandTrendr achieved a similar level of accuracy with CCDC in many aspects, with simpler modeling.

### 5.2. Characteristics and Adaptability of the Three Algorithms

VCT has the advantage of providing an overall picture of the vegetation status of the whole region by analyzing yearly IFZ values (Equations (1) and (2)). The threshold definition based on IFZ can achieve year-by-year monitoring results for each forest pixel in the study area. Furthermore, VCT can refine the forest disturbance monitoring process by combining machine learning [27] or using multiple sources of data for fusion [37]. These refined monitoring results contribute to seeking solutions for many scientific problems, such as accurate forest biomass calculations [17] and forest fragmentation analysis [37,38]. However, when the input Landsat data quality is poor, VCT will not only decrease the quality of VCT monitoring in the current year but also affect the accuracy of the whole time series. Therefore, in this regard, VCT is worse than algorithms that use time series fitting to remove noises or errors, such as LandTrendr and CCDC, both in terms of accuracy and monitoring effectiveness.

The GEE version CCDC uses all available Landsat images to build time series models for each pixel and then extrapolates the model to all the dates of the monitoring period to produce clear Landsat images at any date [14]. Based on the expanded data capacity, a complete model including the monitoring of the seasonal variation can be adopted by adding the parameters of $a_{3,i}$ and $b_{3,i}$ in Equation (3) to characterize the seasonal gradients of forest pixel change, which turns CCDC more acute and delicate [14]. By projecting GEE version CCDC into the monitoring of long time series, it is possible to monitor the subtle changes including greenness trends in the ecosystem layers [39]. In recent years, improved algorithms based on CCDC have emerged, such as the COLD algorithm [40], which makes its fitting method more accurate, the MCCDC method [41], which filters the range of input data, and the S-CCDC method [42], which incorporates a state-space model. Taking advantage of its sharp and progressive advantages, CCDC will be more widely used in forest monitoring.

In fact, CCDC is too sensitive when monitoring forest abrupt changes over long time series, and the cost of retaining the full monitoring results is that the true monitoring patches will be covered by trivial and weak changes, which makes the CCDC results cluttered and fragmented [23]. Instead, LandTrendr simplifies the well-fitting but complex model and does not consider the forest changes within one year. Therefore, LandTrendr does not include seasonal and intra-annual changes, but when monitoring abrupt forest change in long time series, it can obtain more complete patches than CCDC, and the whole effect of abrupt change monitoring is better.

LandTrendr is the most efficient method for forest change monitoring in terms of the time complexity of running, the accuracy of the results, as well as the completeness of the monitored patches. Running LandTrendr algorithm locally is very complex and often takes more than ten hours or even days to complete [43]. The GEE version of LandTrendr compresses the pre-processing steps, calculates them more quickly, and obtains the monitoring results in less than half an hour, faster than both VCT and CCDC running in the same application. It is easier to adjust parameters and filter settings, which greatly improves the efficiency of algorithm and broadens the application areas. In view of such advantages, more and more long time series forest monitoring missions have taken LandTrendr algorithm as the first choice, whether it is applied in integration with other methods such as VCT, EWMACD, MIICA [44], and CCDC [45] or adding deep learning algorithms to further refine the monitoring results [16,46], and these efforts have made LandTrendr a reliable and efficient algorithm suitable for further developments. From a broader perspective, the perfect presentation of GEE version LandTrendr can be expanded not only in the field of forest monitoring but also in other domains related to forest change, such as impervious surface expansion [19], fire-induced forest recovery and renewal [47], and ecological monitoring of open pit mines [48].

### 5.3. Forest Disturbance Monitoring Algorithm Suited to Southeastern China

Coniferous and bamboo forests are widely distributed in Lishui, and plantation forest industries are intensive [49], so forest disturbance events in Lishui are principally driven by human activities, such as clear-cutting and post-harvesting regeneration. Considering the regional characteristics of Lishui, such as developed plantation forest industry, a long history of forest development, fewer natural disasters in a subtropical climate, and well-constructed protected areas, LandTrendr is a more suitable algorithm to monitor forest changes in southeastern China than CCDC and VCT. And LandTrendr can eliminate the influence of cloud and shadow problems on Landsat image quality by yearly selecting images, and it can directly obtain information on the occurrence time of forest disturbance and the intensity of disturbance to support the mapping [16], which is more beneficial for wider ecological applications.

### 6. Conclusions

In this study, based on the Landsat time series observations archived on GEE from 1987 to the present, three algorithms, including LandTrendr, VCT, and CCDC, were implemented to monitor forest abrupt changes in the Lishui of Zhejiang Province from 1990 to 2020, and the similarities and differences among the three methods were compared. The overall spatial accuracies reached 0.852 (LandTrendr), 0.830 (VCT), and 0.862 (CCDC), the kappa coefficients reached 0.704 (LandTrendr), 0.660 (VCT), and 0.727 (CCDC), and the average temporal accuracy reached to 87.30% (LandTrendr), 79.33% (VCT), and 87.74% (CCDC). In conclusion, GEE is a convenient, flexible, and efficient platform that facilitates long-term forest change monitoring in conjunction with highly automated algorithms, streamlines the cost required for downloading and storing data, brings down the application threshold of the monitoring algorithms, and expands their applicability. LandTrendr obtains the most refined results, monitors the most intense and prominent disturbance patches, and neglects some small patches. Therefore, LandTrendr is highly recommended in Southern China or similar regions when monitoring abrupt forest change events.

**Author Contributions:** Conceptualization, M.L.; methodology, M.L. and N.D.; software, N.D.; validation, N.D.; formal analysis, N.D.; investigation, N.D.; resources, N.D.; data curation, N.D.; writing—original draft preparation, N.D.; writing—review and editing, M.L.; visualization, N.D.; supervision, M.L.; project administration, M.L.; funding acquisition, M.L. All authors have read and agreed to the published version of the manuscript.

**Funding:** This research was jointly funded by the Forestry Science and Technology Innovation and Promotion Project Sponsored by Jiangsu Province (LYKJ(2022)02), the National Natural Science

Foundation of China (grant No. 31971577) and the Priority Academic Program Development (PAPD) of Jiangsu Higher Education Institutions.

**Data Availability Statement:** Data are contained within the article.

**Conflicts of Interest:** The authors declare no conflict of interest.

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
