# Peer review of "Mapping Forest Abrupt Disturbance Events in Southeastern China—Comparisons and Tradeoffs of Landsat Time Series Analysis Algorithms"

_remotesensing, doi:10.3390/rs15225408_

Round 1
Reviewer 1 Report
Comments and Suggestions for Authors
1. The content in lines 103-105, "And the main dominant species are horsetail pine (Pinus massoniana Lamb.), Chinese fir (Cunninghamia lanceolata (Lamb.) Hook.) and oak (Cyclobalanopsis glauca (Thunb.) Oerst.)", actually covers only a fraction of the main dominant species.
2. Almost all of the figures are blurry.
3. The description in lines 205-206, "the ??,? with ??,? represents the intra-annual coefficient of variation of reflectance of band ?", is not easy to understand.
4. The representation in lines 213-216, "When the number of consecutive clear images is greater than or equal to 6, the simple model can be fitted, and when the number of consecutive clear images exceeds 24, the complete time series model is estimated by using the LASSO modeling method, the clearer observations, the more complex the model used, the better the fitting effect", is not easy to understand.
5. Lines 222-223 states, "This can unify different detection algorithms into same validation framework, making the accuracy comparable with each other." It is not clear what "This" refers to.
6. The accuracy statistics in lines 248-249, including "overall spatial accuracy, kappa coefficients, user’s accuracy and producer’s accuracy", need to be described with formulas in the text.
7. Tables 1, 2, and 3: Why are "user’s accuracy" and "producer’s accuracy" calculated separately for "forest change" and "non-change" cases, while "overall accuracy" and "kappa" are calculated together? Also, "user’s accuracy" and "producer’s accuracy" are expressed in percentages, while "overall accuracy" is in decimals. A unified percentage representation should be used.
8. The first letter of "total" in Tables 1, 2, and 3 should be capitalized.
9. How is the least omission error in line 258, "least omission error of 8.10% (1.00 - 91.90%)", calculated?
10. Lines 281-282: "the temporal accuracy of CCDC and Landtrendr was better than that of VCT" should use the plural form for "accuracy".
11. Figure 5: Forest abrupt changes mapping from 1990 to 2020 in Lishui derived from VCT, Landtrendr and CCDC algorithms. I have a question: If the same plot experienced multiple changes from 1990 to 2020, the previous changes would be covered by the last change. How can you represent the intermediate state changes? Figure 6 also has a similar problem.
12. Line 330: "Obviously, the morphology of patches detected by" should use the plural form for "morphology". Line 337: "Landtrendr’s performance was both accurate and clear" also involves confusion between singular and plural usage.
13. The representation in lines 380-381, "for determining the forest changes cause some commission and omission errors in the monitoring process", needs to be phrased more clearly.
14. The description in lines 472-473, "which filters the range of input data, and the S-CCDC method[41], which incorporates a state-space model.", seems unclear.
15. Should a “.” be added at the end of the 5-th reference in line 550?
Comments on the Quality of English Language
1. The content in lines 103-105, "And the main dominant species are horsetail pine (Pinus massoniana Lamb.), Chinese fir (Cunninghamia lanceolata (Lamb.) Hook.) and oak (Cyclobalanopsis glauca (Thunb.) Oerst.)", actually covers only a fraction of the main dominant species.
2. Almost all of the figures are blurry.
3. The description in lines 205-206, "the ??,? with ??,? represents the intra-annual coefficient of variation of reflectance of band ?", is not easy to understand.
4. The representation in lines 213-216, "When the number of consecutive clear images is greater than or equal to 6, the simple model can be fitted, and when the number of consecutive clear images exceeds 24, the complete time series model is estimated by using the LASSO modeling method, the clearer observations, the more complex the model used, the better the fitting effect", is not easy to understand.
5. Lines 222-223 states, "This can unify different detection algorithms into same validation framework, making the accuracy comparable with each other." It is not clear what "This" refers to.
6. The accuracy statistics in lines 248-249, including "overall spatial accuracy, kappa coefficients, user’s accuracy and producer’s accuracy", need to be described with formulas in the text.
7. Tables 1, 2, and 3: Why are "user’s accuracy" and "producer’s accuracy" calculated separately for "forest change" and "non-change" cases, while "overall accuracy" and "kappa" are calculated together? Also, "user’s accuracy" and "producer’s accuracy" are expressed in percentages, while "overall accuracy" is in decimals. A unified percentage representation should be used.
8. The first letter of "total" in Tables 1, 2, and 3 should be capitalized.
9. How is the least omission error in line 258, "least omission error of 8.10% (1.00 - 91.90%)", calculated?
10. Lines 281-282: "the temporal accuracy of CCDC and Landtrendr was better than that of VCT" should use the plural form for "accuracy".
11. Figure 5: Forest abrupt changes mapping from 1990 to 2020 in Lishui derived from VCT, Landtrendr and CCDC algorithms. I have a question: If the same plot experienced multiple changes from 1990 to 2020, the previous changes would be covered by the last change. How can you represent the intermediate state changes? Figure 6 also has a similar problem.
12. Line 330: "Obviously, the morphology of patches detected by" should use the plural form for "morphology". Line 337: "Landtrendr’s performance was both accurate and clear" also involves confusion between singular and plural usage.
13. The representation in lines 380-381, "for determining the forest changes cause some commission and omission errors in the monitoring process", needs to be phrased more clearly.
14. The description in lines 472-473, "which filters the range of input data, and the S-CCDC method[41], which incorporates a state-space model.", seems unclear.
15. Should a “.” be added at the end of the 5-th reference in line 550?
Reviewer 2 Report
Comments and Suggestions for Authors
1.In the introduction, I do not seem to see the focus of this study and the scientific problems that need to be solved. Please elaborate.
2.L86-94 should be included in the profile of the study area. What is the purpose of introducing the basic situation of the study area in the introduction?
3.The figures in the paper are all not clear, and it is difficult to distinguish the specific content that the author wants to express, which brings great trouble to reading.
4.L156-162, what are the specific bands used ? I hope to make a detailed explanation, otherwise it will bring trouble to readers who do not understand the bands information.
5. Does L224-227 belong to the introduction?
6.I suggested to adjust the structure of the results and discussion of the article. Firstly, the results part elaborates the results obtained from the figure or table, briefly expounds the obvious results, and puts the inferences obtained from these obvious results in the results into the discussion ; secondly, in the discussion, the relevant inferences are discussed in detail around the results obtained in this paper, and the relevant results of this paper are discussed in detail.
Comments on the Quality of English LanguageSimple modifications to complex languages.
Reviewer 3 Report
Comments and Suggestions for Authors
1. Lishui city, Zhejiang is characterized by extensive plantation forests, regular forest logging and post-harvesting forest recovery practices as well as casual forest fire events. Because traditional remote sensing-based forest disturbance is inefficient in the long-time span, this study, based on the GEE platform, used three algorithmics including VCT, LandTrendr and CCDC to monitor forest abrupt changes in the Lishui from 1990 to 2020.
2. The description of trade-off factors for three algorithmics in the implementation process is very helpful to understand their similarities and differences. Each algorithmic has advantage on its expertise area with limitation on some operations.
3. By means of kappa analysis and deeply discussion, this study showed that Landtrendr is a more suitable algorithm to monitor forest changes in southeastern China than CCDC and VCT.
4. The words of “LandTrendr” and “Landtrendr” were used simultaneously in this study. Please use one of them in this study.
Round 2
Reviewer 1 Report
Comments and Suggestions for Authors
1.Whether the clarity of the image meets the publication requirements needs further comprehensive inspection.
2.I think there is a need for further refinement in the English expression.
Comments on the Quality of English Language1.Whether the clarity of the image meets the publication requirements needs further comprehensive inspection.
2.I think there is a need for further refinement in the English expression.
Reviewer 2 Report
Comments and Suggestions for Authors
The manuscript was upgraded after revision.
